# Ectopic Pregnancy and T-Cell Lymphoma in a Eurasian Red Squirrel (*Sciurus vulgaris*): Possible Comorbidity and a Comparative Pathology Perspective

**DOI:** 10.3390/ani14050731

**Published:** 2024-02-27

**Authors:** Caterina Raso, Valentina Galietta, Claudia Eleni, Marco Innocenti, Niccolò Fonti, Tiziana Palmerini, Mauro Grillo, Pietro Calderini, Elena Borgogni

**Affiliations:** 1Istituto Zooprofilattico Sperimentale del Lazio e Della Toscana “M. Aleandri”, 02100 Rieti, Italy; pietro.calderini@izslt.it (P.C.); elena.borgogni@izslt.it (E.B.); 2Istituto Zooprofilattico Sperimentale del Lazio e Della Toscana “M. Aleandri”, 00178 Roma, Italy; valentina.galietta@izslt.it (V.G.); claudia.eleni@izslt.it (C.E.); tiziana.palmerini@izslt.it (T.P.); 3UOC Igiene e Sanità Animale, ASL Rieti, 02100 Rieti, Italy; m.innocenti@asl.rieti.it (M.I.); m.grillo@asl.rieti.it (M.G.); 4Dipartimento di Scienze Veterinarie, Università di Pisa, Viale delle Piagge, 56124 Pisa, Italy; niccolo.fonti@phd.unipi.it

**Keywords:** *Sciurus vulgaris*, ectopic pregnancy, T-cell lymphoma

## Abstract

**Simple Summary:**

*Sciurus vulgaris*, commonly known as the Eurasian red squirrel, is a wild rodent species widely distributed throughout Europe and currently classified as “least concern” by the International Union for Conservation of Nature (IUCN). Despite its conservation status, several threats, such as infectious diseases, habitat fragmentation and the introduction of the alien invasive species grey squirrel (*Sciurus carolinensis)*, could affect *S. vulgaris* populations. Ectopic pregnancy is a condition consisting of a pregnancy developing outside the uterus. Tumors affecting the reproductive tract are known to predispose to this condition. In this paper, we describe for the first time a case of lymphoma leading to an ectopic pregnancy in a Eurasian red squirrel. A deeper knowledge of the pathology of this species is important for estimating the impact of diseases on the *S. vulgaris* population. Moreover, wildlife can be a sentinel for pathologies that have environmental causes and that could affect other animals and humans that share the same environment. Thus, deeper knowledge of wildlife cancer incidence and the environmental and individual causes of cancer development is relevant from a comparative pathology perspective.

**Abstract:**

Ectopic pregnancy (EP) is a life-threatening disease that affects humans and other mammals. Tumors causing ruptures of the reproductive tract have been identified as possible predisposing factors in human and veterinary medicine. We here describe a case of concomitant ectopic pregnancy and lymphoma in a Eurasian red squirrel found deceased in Italy and submitted to the public health laboratory Istituto Zooprofilattico Sperimentale del Lazio e della Toscana (IZSLT) for post-mortem examination. A full-term partially mummified ectopic fetus in the abdomen and a large fibrinonecrotic tubal scar adjacent to the right ovary were observed at necropsy. The tubal scar is likely the point of tubal rupture through which the fetus displaced. Histology revealed the presence of neoplastic cells referable to lymphoma infiltrating the ovary, spleen, small intestine, heart and peripancreatic adipose tissue. The lymphoma was further characterized as T-cell-type using immunohistochemistry. We suggest that the lymphoma, by involving the ovary, played a pathogenetic role in the development of a secondary EP by altering the genital tract at the structural and hormonal levels. To the best of our knowledge, this is the first report of concomitant ovarian lymphoma and EP in animals and humans in the literature.

## 1. Introduction

The Eurasian red squirrel *(Sciurus vulgaris)* is a rodent species with a large geographic and habitat range, which includes forests and urban environments from Europe to Asia. According to the International Union for Conservation of Nature (IUCN) [1], *S. vulgaris* falls under the Least Concern (LC) category. Despite its wide geographical distribution, several threats could impair its conservation status, such as emerging infectious diseases, habitat fragmentation and the introduction of the alien invasive eastern grey squirrel species (*Sciurus carolinensis)*, an important competitor for food and habitat [1]. Eurasian red squirrel offspring are born in February–April and May–August, with one to six individuals per litter [1,2] after a mean gestation length of 37.44 days [3]. Sciuridae have a hemochorial discoid placenta, similar to other rodents and primates, and a duplex uterus [4,5].

We here describe a case of concomitant occurrence of ectopic pregnancy (EP) and T-cell lymphoma in a Eurasian red squirrel found deceased in Italy. EP is a well-known life-threatening disease that affects humans and, although less frequently, other mammals [6]. It has been described in domestic [7,8,9], laboratory [10,11,12,13] and wild animals [14,15,16]. Two main EP types are recognized, tubal and abdominal pregnancy, depending on whether the gestation develops in the fallopian tubes or in the abdominal cavity, respectively. The etiopathogenesis of EP in animals has not been completely clarified. Several predisposing factors have been identified, including iatrogenic factors and surgical procedures [9,11,17], hormonal factors altering the normal ovum transport and implantation [6] and tumors and other spontaneous reproductive pathologies causing ruptures of the reproductive tract [15,18,19,20,21].

Lymphoma comprises a diverse group of hematopoietic malignancies that arise from the clonal proliferation of lymphocytes not only in the hematolymphoid system but also in extranodal sites [22]. It is one of the most frequent tumors in domestic animals [22,23] and laboratory rodents [24]. Several causes, such as individual factors, retroviruses, chemicals and irradiation, are involved in the etiology of lymphoma in rodents [24]. Oncogenesis results from the interplay between endogenous and exogenous genomic stressors: chronic inflammation, immune system, variation in cell biology between species and individuals, pathogen infection and environmental pollutants [25,26]. In veterinary medicine, most epitheliotropic lymphomas in the skin and the small intestine have a T-cell origin [23]. Moreover, most of the retrovirus-associated lymphomas of cats, mice and humans in the literature display a T-cell immunophenotype [23,24]. T-cell lymphoma has also been experimentally induced in genetically engineered mice using chemicals [24]. There are descriptions of multicentric and epitheliotropic lymphoma in red squirrels in the literature [27,28] and one record of a suspected EP [14], but no cases of concomitant ectopic pregnancy and lymphoma have been described, neither in *S. vulgaris* nor in other species or humans in the literature. This case report aims to unravel the possible comorbidity between the two conditions.

## 2. Materials and Methods

On 3 April 2023, an adult, female, free-ranging Eurasian red squirrel was found dead in Italy (42.41273700, 12.88127000) by the veterinary officers of the local health authority ASL of Rieti (Latium region, Central Italy). The day after its retrieval, the carcass was submitted to the territorial division of the public health laboratory Istituto Zooprofilattico Sperimentale del Lazio e della Toscana and underwent post-mortem examination. The body condition was estimated according to the score proposed by Ullman-Culleré and Folz for laboratory mice [29]. Samples of the ovaries, uterus, lung, heart, liver, spleen, kidney, intestine, pancreas and all the fetal organs were collected during the necropsy, fixed in 10% buffered formalin and routinely processed for histopathological examination. To further characterize the neoplasm, immunohistochemistry with polyclonal rabbit anti-human CD3 (Dako) and polyclonal rabbit anti-human CD20 (Thermo Fisher Scientific, Waltham, MA, USA) was performed on unstained sections of the affected organs using the labeled streptavidin–biotin (LSAB) peroxidase method.

## 3. Results

The animal was in suboptimal body condition (BCS 2/5 [29]). Necropsy revealed the presence of a full-term fetus in the abdominal cavity. It measured 5 cm in crown–rump length, was early mummified, covered in a thin smooth yellow-grey serosal membrane, enveloped in the peritoneum and connected to the omentum through a thin fibrovascular band (Figure 1a). The fetal organs were almost fully developed, consistent with a late gestational stage, appeared autolytic and did not show macroscopically evident pathological findings. Gross examination of the adult squirrel’s reproductive tract revealed moderate hyperemia of the right uterine horn wall and the presence of a soft reddish spherical mass measuring 1.5 cm, topographically adjacent to the right ovary and involving the tubal infundibulum wall (Figure 1b,c). A viable placenta was not present, neither enveloping the abdominal fetus nor in the uterus. Severe diffuse splenomegaly was observed, with symmetric enlargement and rounded edges of the organ. Other gross pathologic findings were severe segmental thickening of the jejunal wall, moderate diffuse catarrhal enteritis and enterocolitis and severe hyperemia of the corresponding serosa. Microscopically, a dense, not delimited, non-capsulated, infiltrative, round cell neoplasm referable to lymphoma was observed in the right ovary (Figure 2a), spleen (Appendix A) and small intestine (Figure 2b). Neoplastic cells were also visible in the peripancreatic adipose tissue (Appendix A) and in the small lymphatic vessels of the heart (Appendix A). Lymphoid cells had a nest-like arrangement in the ovarian parenchyma, while they diffusely infiltrated the other affected tissues. They were round to oval, with scant weakly basophilic cytoplasm. Nuclei were round to rarely indented, measured 1.5–2 times the volume of a red blood cell and showed peripheralized chromatin and one prominent basophilic nucleolus. Cellular pleomorphism was moderate, and the mitotic count ranged from two to four in 2.37 mm^2^. Neoplastic cells stained positive for CD3 and negative for CD20 at the IHC, consistent with T-cell lymphoma (Figure 2c). Scattered lymphocytes expressing CD20 (B-cells) were present and attributed to secondary inflammation (Figure 2d). The periovarian spherical lesion was composed of fibrin, necrosis and erythrocytes and was consistent with a scar at the tubal level. Histologically, the fetal organs were altered by autolytic changes referable to mummification, thus preventing a thorough microscopic evaluation of the fetus.

## 4. Discussion

The EP described in this case is referable to an abdominal pregnancy, similar to most of the diagnosed spontaneous ectopic pregnancies of mammals other than primates reported in the literature [6,8,10,12,13,16]. Based on the pathogenesis, abdominal pregnancy may be further subdivided into primary and secondary subtypes. In the primary subtype, fertilization or the first implantation of a fertilized oocyte occurs in the abdominal cavity, with placentation developing either on a peritoneal or an omental surface. In the secondary subtype, the fetus is expelled into the peritoneal cavity after implantation, following a rupture of the oviduct or uterus [6]. We can only hypothesize the underlying pathogenesis of the specific case reported in this paper. The ovarian neoplastic lesion altered the tubal structure at the periovarian level, predisposing to a rupture, followed by the displacement of the developing fetus from the gravid uterine horn to the abdominal cavity. Moreover, the ovarian neoplasm, by infiltrating the functional ovarian tissue, may have impaired the hormonal function and altered tubal motility, thus concurring the EP development [6,30]. Thus, our hypothesis is consistent with a secondary EP subtype: the fibrinonecrotic scar was likely the point of tubal rupture, and the ovarian lymphoma played a role in the development of the EP. Complete reimplantation of the placenta in the abdominal cavity or an internal abortion may occur following the loss of placental attachment caused by fetal escape into the abdominal cavity. We could not evaluate placental features as a viable placenta was not observable in our case, consistent with most of the records on abdominal pregnancy [6].

Although several cases of concomitant genital tract neoplasia and ectopic pregnancy have been described in human [18,19,20,21] and veterinary medicine [15], to our knowledge, no cases of concomitant lymphoma and ectopic pregnancy have been reported so far. The neoplasia described in this paper is a T-cell lymphoma involving the ovary, spleen, intestine, heart and peripancreatic tissue. In the literature, one case of multicentric lymphoma was recorded but not immunophenotyped [27], whereas one case of epitheliotropic lymphoma was characterized as T-cell in Eurasian red squirrels [28]. In our case, it was not possible to assess whether the lymphoma’s primary site was the spleen, the jejunal Peyer plaques or the ovary or if it arose as a multicentric form.

Lymphomas are known to affect the ovarian tissue of domestic [31] and laboratory animals [32]. As in our case, ovarian lymphoma does not usually cause evident ovarian enlargement [33]. In cattle and humans, neoplastic foci were observed to develop within the corpora lutea [31,34], probably from the lymphocytes that physiologically surround the corpus luteum [33]. This pathogenetic process cannot be excluded in our case since a pregnancy corpus luteum was present and was infiltrated by the neoplastic cells. During pregnancy, lymphoma usually displays placental metastasis, leading to pregnancy complications and stillborn fetuses [35], but the absence of the placenta and fetal mummification prevented the evaluation of possible placental and fetal metastasis in this report.

Most of the oncogenetic mechanisms leading to the development of lymphoma are still unclear and hard to assess [23]. Interestingly, retroviruses are a possible cause of T-cell lymphoma in rodents [24]. We did not perform tests to exclude retroviruses in the reported case, but the possible correlation between retrovirus and lymphoma deserves further attention.

Examining tumors in several animal species sharing the same environment, including wildlife, can reveal species-specific mechanisms of cancer formation and cancer resistance and different responses to the same environmental stressors in different species [25,26]. Thus, synanthropic wildlife within a particular ecosystem can potentially play the role of both sentinels of environmental carcinogens and models of cancer development [26].

Knowledge of the health status of the majority of wildlife species is limited in most European countries [36]. As cancer in wildlife is mostly diagnosed by necropsy [37], the setup of networks for the detection of carcasses and systematic post-mortem examination of free-ranging species is crucial for understanding the impact of diseases, including cancer, and must be considered as a pillar in wildlife health surveillance programs.

## 5. Conclusions

To the best of our knowledge, this is the first reported case of concomitant lymphoma and secondary abdominal pregnancy. Our findings suggest that the ovarian lymphoma played a role in the development of the EP by altering the structure of the genital tract. By involving and altering the ovarian parenchyma, we suggest that the tumor could have also impaired the hormonal function of the corpus luteum in the maintenance of pregnancy, although this could not be confirmed by serial hormonal assays in the alive animal. Little is known about Eurasian red squirrel reproductive pathology and oncology, despite its affinity with other better-known rodents. Further studies are needed to estimate the incidence of lymphoma in red squirrels, concomitance with ectopic pregnancy and factors involved in tumorigenesis. A deeper knowledge of ecopathology is crucial for understanding the real impact of diseases on wildlife species from a conservation medicine perspective. Moreover, further studies are needed to clarify the environmental drivers of oncogenesis in the Eurasian red squirrel and other free-ranging wild species. The development of wildlife cancer registries and integration with other human and animal information systems would expand our knowledge on tumorigenesis and eventually help in identifying possible sentinels for environmental risk factors from a One Health perspective.

## Figures and Tables

**Figure 1 animals-14-00731-f001:**
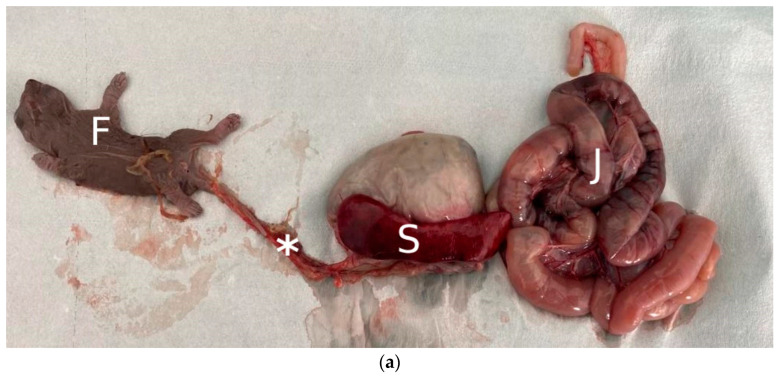
Gross findings at necropsy: (**a**) a thin fibrovascular cord (asterisk) connecting the full-term fetus (F) to the omentum; evident splenomegaly (S) and segmental enteropathy of the jejunum (J). (**b**) Genitourinary tract, gross: round lesion (black arrowhead) of 1.5 cm in diameter adjacent to the right ovary and hyperemia of the right uterine horn (black arrow). (**c**) Fibrin and necrotic-hemorrhagic tissue were evident on the cross-section of the periovarian tubal lesion.

**Figure 2 animals-14-00731-f002:**
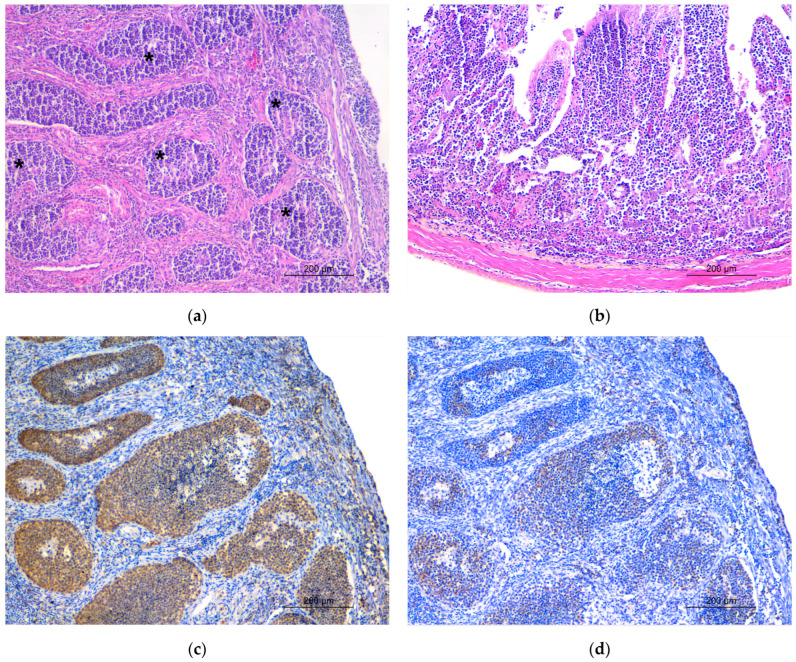
Histological appearance of lymphoma in hematoxylin and eosin-stained sections: (**a**) neoplastic cells are arranged in irregular nest-like structures in the ovary (asterisk) and (**b**) diffusely infiltrate the lamina propria of the jejunum. Immunohistochemistry, ovary: neoplastic cells stain (**c**) positive (brown color) for CD3 and (**d**) negative for CD20 markers. (**d**) Scattered cells staining positive for CD20 are referable to inflammatory lymphocytes.

## Data Availability

The original contributions presented in this study are included in this article/Appendix A. Further inquiries can be directed to the corresponding author.

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
