# Peer review of "Ectopic Pregnancy and T-Cell Lymphoma in a Eurasian Red Squirrel (*Sciurus vulgaris*): Possible Comorbidity and a Comparative Pathology Perspective"

_animals, 2024, doi:10.3390/ani14050731_

Round 1
Reviewer 1 Report
Comments and Suggestions for Authors
The present investigation deals with the case report of "Ectopic pregnancy and T-cell lymphoma in a Eurasian red squirrel (Sciurus vulgaris)". This research provides interesting information. However, it is advisable to make some changes before final publication.
INTRODUCTION
Line 57.- "Ectopic pregnancy (EP)", however "EP" was already defined in line 28. I recommend only using abbreviations. Check if abbreviations are used in this section.
Line 67-69.- "Lymphoma comprises a diverse group of hematopoietic malignancies that arise from the clonal proliferation of lymphocytes not only in the hematolymphoid system but also in extranodal sites". I recommend mentioning the risk factors for T-cell lymphoma and a more detailed description of this pathology.
CASE DESCRIPTION
Line 81-82.- "The animal was in suboptimal body condition". Isn't there a more accurate scale in the literature to determine body condition?
Fig 1. They mention "Spenomegaly" and "segmental enteropathy of the jejunum", however, the paper does not mention this.
Fig 2. They mention "negative for CD20 markers". But in the excision on line 106 they mention “Scattered lymphocytes expressing CD-20 (B-cells) were present and attributed to secondary inflammation (Figure 2d)”.
DISCUSSION
In general, I recommend describing the pathways or mechanisms by which ED and T-cell lymphoma may be related. In addition, try to restructure this section according to how the clinical findings were previously described in the previous section "CASE DESCRIPTION" and then try to explain how this clinical picture could have been presented.
Line 120.- “The ectopic pregnancy”. I recommend using the abbreviation "EP" for this pathology. This is because it has already been abbreviated previously.
CONCLUSION
I recommend restructuring this section and being more specific about the findings mentioned in "CASE DESCRIPTION and DISCUSSION". In addition, they mention in line 171 "hormonal level" nowhere in their findings do they mention hormonal levels so they cannot consider this. It would be speculative; I recommend eliminating it.
Author Response
Thank you very much for taking the time to review this manuscript.
According with your suggestions we modified our manuscript as follows:
- only used the EP abbreviation, after defining it in the introduction;
- provided a more detailed decription of T-cell lymphoma pathology and risk factors in the introduction;
- clarifyed the score used for assessing the body condition and cited it in the references;
- better described macroscopic findings in the results;
- clarified and better described the immunohistochemistry results under figure 2d;
- better described the link between ectopic pregnancy and lymphoma in the discussion;
- suggested how the neoplastic involvement of the ovary, and consequently, of the corpus luteum, could have impaired the hormonal function. As suggested, we did not mention the hormonal levels since hormonal analysis on the alive animal could not be performed.
We hope to have complied with the suggestions. Please see the attachment and the updated manuscript for more details.

Reviewer 2 Report
Comments and Suggestions for Authors
The case report is interesting. However, there are many weaknesses that need improvement.
Hypothesis:
1. a neoplastic lesion in the ovarian region would, in my opinion, predispose to the development of a primary ectopic pregnancy. The fertilised ovum entered the abdominal cavity and implanted in the omentum (as described in the literature). In addition, the fibrovascular cord may be a manifestation of the umbilical cord which is a combination of the fetus and the placenta (which most often develops in the omentum). Please think about this and respond.
M&M
1. Why isn't the M&M chapter separated????TIt's This is unacceptable and unreadable! please correct it!
Description:
1. the case description should be better structured. In the results, please describe separately the changes in each organ as well as the fetus.
2. please add more precise dimensions of the fetus (weight, estimated day of development). A big disadvantage is the lack of description of the internal organs of the fetus.
3. an examination of the placenta is required to distinguish primary from secondary ectopic pregnancy - please provide information and photographs
4. histological preparations of the corpus luteum, which are, as it were, a manifestation of maternal and fetal cooperation, will also be required.
Discussion:
1. As can be seen from the photograph, the fetus is at an advanced stage of development. It is therefore a significant error for me not to inform and discuss the presence or absence of the placenta. As is known from the literature, in secondary pregnancies due to rupture of the fallopian tube (as suggested by the researchers) there is a reduction in the blood supply to the placenta and the fetus dies at an early stage. However, there are situations in which the fetus can develop to an advanced stage. Because of the possibility of these two situations, I require that information on the placenta be presented and discussed in the discussion.
2. I ask that the placenta be better discussed in the discussion. A big stretch is the information about the placenta in the 150-152nd verse.
Author Response
Thank you very much for taking the time to review this manuscript. Please see the attachment for more details.

Reviewer 3 Report
Comments and Suggestions for Authors
line 56-57: We here describe a case of concomitant occurrence of ectopic pregnancy (EP) and T-cell lymphoma in a 56 Eurasian red squirrel found deceased in Italy. Ectopic pregnancy (EP) is a well-known 57 life-threatening disease that affects humans and, although less frequently, other mammals;
line 116-118: expand the description of the images in the caption and indicate the cells with arrows or asterisks
Have any analyses been performed on the mummified fetus? Or the fetus was to much mummified in order to allow any hystological examination to evaluate a possible maternal-fetal transmission of lymphoma?
Author Response
Thank you very much for taking the time to review this manuscript. Please see the attachment.

Round 2
Reviewer 2 Report
Comments and Suggestions for Authors
Thank you for your responses. I accept the manuscript in its current form.
I wish you the best of luck with your further work. Best Regards